# Importance, Applications and Features of Assays Measuring SARS-CoV-2 Neutralizing Antibodies

**DOI:** 10.3390/ijms24065352

**Published:** 2023-03-10

**Authors:** Pia Gattinger, Anna Ohradanova-Repic, Rudolf Valenta

**Affiliations:** 1Department of Pathophysiology and Allergy Research, Center for Pathophysiology, Infectiology and Immunology, Medical University of Vienna, 1090 Vienna, Austria; 2Institute for Hygiene and Applied Immunology, Center for Pathophysiology, Infectiology and Immunology, Medical University of Vienna, 1090 Vienna, Austria; 3Karl Landsteiner University, 3500 Krems an der Donau, Austria; 4Laboratory for Immunopathology, Department of Clinical Immunology and Allergology, Sechenov First Moscow State Medical University, 119435 Moscow, Russia; 5NRC Institute of Immunology FMBA of Russia, 115478 Moscow, Russia

**Keywords:** SARS-CoV-2, serological assays, molecular assays, neutralizing antibodies

## Abstract

More than three years ago, the Severe Acute Respiratory Syndrome Coronavirus 2 (SARS-CoV-2) caused the unforeseen COVID-19 pandemic with millions of deaths. In the meantime, SARS-CoV-2 has become endemic and is now part of the repertoire of viruses causing seasonal severe respiratory infections. Due to several factors, among them the development of SARS-CoV-2 immunity through natural infection, vaccination and the current dominance of seemingly less pathogenic strains belonging to the omicron lineage, the COVID-19 situation has stabilized. However, several challenges remain and the possible new occurrence of highly pathogenic variants remains a threat. Here we review the development, features and importance of assays measuring SARS-CoV-2 neutralizing antibodies (NAbs). In particular we focus on in vitro infection assays and molecular interaction assays studying the binding of the receptor binding domain (RBD) with its cognate cellular receptor ACE2. These assays, but not the measurement of SARS-CoV-2-specific antibodies per se, can inform us of whether antibodies produced by convalescent or vaccinated subjects may protect against the infection and thus have the potential to predict the risk of becoming newly infected. This information is extremely important given the fact that a considerable number of subjects, in particular vulnerable persons, respond poorly to the vaccination with the production of neutralizing antibodies. Furthermore, these assays allow to determine and evaluate the virus-neutralizing capacity of antibodies induced by vaccines and administration of plasma-, immunoglobulin preparations, monoclonal antibodies, ACE2 variants or synthetic compounds to be used for therapy of COVID-19 and assist in the preclinical evaluation of vaccines. Both types of assays can be relatively quickly adapted to newly emerging virus variants to inform us about the magnitude of cross-neutralization, which may even allow us to estimate the risk of becoming infected by newly appearing virus variants. Given the paramount importance of the infection and interaction assays we discuss their specific features, possible advantages and disadvantages, technical aspects and not yet fully resolved issues, such as cut-off levels predicting the degree of in vivo protection.

## 1. Introduction

The COVID-19 pandemic, which started in late 2019 and was caused by the Severe Acute Respiratory Syndrome Coronavirus 2 (SARS-CoV-2) [1,2], has led to millions of deaths worldwide. Interestingly, the characteristics of the COVID-19 pandemic have changed in the recent past. Since 2022, we have observed high infection rates, but the number of COVID-19-associated deaths has declined (https://coronavirus.jhu.edu/map.html) (accessed on 18 January 2023). Several factors may contribute to this improvement of the situation. In fact, by now the majority of the population has been infected and/or vaccinated and thus has built up a basic SARS-CoV-2-specific immune response. Furthermore, currently less pathogenic virus variants of the omicron lineage are dominant. However, a considerable proportion of the population, in particular immunocompromised subjects and vulnerable patients, have difficulties in producing antibodies that protect from new infections [3,4,5,6,7,8]. Furthermore, RNA viruses, such as coronaviruses, influenza viruses and HIV, have extremely high mutation rates due to their replication mechanism and the lack of viral RNA polymerase proofreading activity and accordingly there is a high risk that new highly pathogenic SARS-CoV-2 variants may emerge [9,10]. Therefore, there is a high need for assays that can be used to test if subjects have antibodies, which can protect them by preventing infection with currently prevailing or newly emerging SARS-CoV-2 variants. In this review, we discuss the two main types of assays which can be used to determine if antibodies can protect against infection by SARS-CoV-2, namely virus-neutralization assays and assays studying if antibodies can prevent the binding of the virus receptor binding domain (RBD) of the Spike (S) protein to its receptor ACE2.

The latter assays have been developed in the beginning of the COVID-19 pandemic to answer several important questions, among them whether natural infection would induce a protective immunity in convalescent subjects and for how long the neutralizing antibody response would prevail. Based on the knowledge that the SARS-CoV-2 virus infects the host cells by docking via a portion of the viral S protein termed RBD to its cognate receptor ACE2 on the host cells [11], molecular interaction assays were developed which allowed us to investigate if antibodies from convalescent subjects can inhibit the RBD–ACE2 interaction and thus have virus-neutralizing capacity [12,13,14]. In fact, results from the molecular interaction assays correlated very well with results obtained by classical virus-neutralization tests [15,16,17,18].

Already, the first results obtained with molecular interaction assays in 2020 indicated that a considerable percentage of infected subjects developed insufficiently blocking antibodies preventing the binding of RBD to ACE2 [13]. In agreement with these results, it was observed that many previously infected subjects became infected again a few months later at the end of 2020, and it became clear that infections with SARS-CoV-2 will occur again and again, as is observed for other seasonal respiratory virus infections, of which some are known to also trigger asthma exacerbations, such as rhinovirus (RV) infections, infections by respiratory syncytial virus (RSV) and influenza [19,20,21,22,23].

With the availability of the first COVID-19 vaccines in the beginning of 2021, it was hoped that vaccination would protect vaccinated subjects in a sustainable manner. The vast majority of FDA- and EMA-authorized COVID-19 vaccines are vector- (Vaxzevria, Janssen COVID-19 vaccine) or mRNA-based (Comirnaty, Spikevax) [24,25] (https://covid19.trackvaccines.org/agency/who/) (accessed on 18 January 2023) but more recently adjuvanted protein vaccines (e.g., Novavax COVID-19 vaccine) [26] and vaccines based on inactivated virus have also become available [27]. The available vaccines induce SARS-CoV-2-specific cellular and antibody responses but a major goal is to induce virus-neutralizing antibodies against the S protein and in particular against the RBD of SARS-CoV-2 [28,29]. Similarly to what is observed after natural infection, it was found that COVID-19 vaccines induce a heterogeneous S-specific antibody response regarding antibody titers and virus neutralizing capacity [15,30,31]. Importantly, neutralizing antibodies induced by infection as well as by vaccination were found to decrease after a few months [32,33,34,35], and accordingly breakthrough infections were noted even after booster vaccinations [36,37,38].

While in the beginning of the COVID-19 pandemic a major goal was to achieve a basic SARS-CoV-2-specific immunity through vaccination, there are different challenges and questions now. To name a few of them:Who should receive a booster vaccination, and when? This question has not yet been answered.Who, in particular who of the vulnerable subjects, has developed sufficiently high levels of neutralizing antibodies to be protected against a new infection? It has become clear that there is a high variation in protective antibody responses among such subjects and no generally valid answer is available.What are the reasons why certain subjects do not develop neutralizing antibodies and can they be protected by next-generation COVID-19 vaccines or passive immunization approaches? It is possible that genetic factors determining immune response are responsible for this phenomenon but definitive answers are lacking.Are those who were infected and/or vaccinated protected against infections caused by newly emerging variants? Protection declines over time and it seems to be not possible to predict protection against new variants with certainty at the moment.What biomarkers can we use to predict if somebody is protected against infection? In this article we argue that assays measuring neutralizing antibodies against existing and emerging SARS-CoV-2 variants may be helpful for answering the aforementioned questions. We also highlight the need for further development of these assays to render them useful to predict the extent of in vivo protection, and the need for eventual mass screening at affordable costs.

## 2. Can the Measurement of SARS-CoV-2-Specific Antibodies Predict Virus Neutralization?

Serological assays to determine the SARS-CoV-2-specific immunoglobulin response were developed early in the pandemic to determine seroconversion after infection [39]. It turned out that these assays, which measured antibodies specific to the SARS-CoV-2 nucleocapsid-protein (N), were very useful to indicate past exposure to SARS-CoV-2, but it is obvious that N-specific antibodies were not directly involved in virus neutralization [40,41]. The N protein is a structural protein of SARS-CoV-2, which resides inside the virion and hence does not play a role in the infection process. However, recently it has been demonstrated that the N protein can appear on the surface of infected cells and accordingly one should not exclude the possibility that N-specific immune responses (e.g., CD8^+^ cell responses, antibody-dependent cellular cytotoxicity) may contribute to the elimination of virus-infected cells [42].

By contrast, antibody responses towards the S protein and in particular against RBD, were considered as surrogate markers for a protective antibody response [43] and the majority of neutralizing antibodies target the RBD [15]. Most of the COVID-19 vaccines are based on the S protein of SARS-CoV-2 [29] and only a few vaccines based on inactivated whole virions (e.g., CoviVac) contain also the N protein. Therefore it is possible to discriminate between vaccination and infection by the measurement of S-, RBD- and N-specific antibodies. S- and RBD-specific antibodies will be induced or boosted after vaccination and infection, whereas N-specific antibodies will be induced or boosted only after infection, unless a N-containing vaccine is used. The use of chips containing micro-arrayed SARS-CoV-2 antigens in different conformations, namely as folded and unfolded proteins and S-derived synthetic peptides lacking a structural fold for the analysis of SARS-CoV-2-specific antibodies after infection and vaccination, has revealed that antibodies directed against folded S and especially against folded RBD are associated best with virus-neutralization [15,44]. Relatively small increases in antibodies specific to S-derived peptides have been noted after infection and after vaccination, indicating that not only antibodies directed to folded RBD but also antibodies with other specificities may contribute to virus neutralization [15,45,46,47,48,49], which needs to be further investigated.

Multiplex assays containing several SARS-CoV-2 antigens [15,44,50] will therefore be useful to discriminate between different types and specificities of SARS-CoV-2-specific antibodies and provide different diagnostic information within a single test. Since antibodies do not solely have virus neutralizing capacity but also have the ability to opsonize the virus and subsequently activate complement or induce antibody-dependent cytotoxicity [51], performing quantitative measurements of SARS-CoV-2-specific antibody levels and correlating them with in vivo protection determined by virus neutralization has been attempted [52]. However, a quantitative cut-off value which is thought to protect from reinfection has not been known until now. At present, it seems that antibody levels directed against folded RBD reflect the ability of antibodies to neutralize SARS-CoV-2 best but no cut-off levels and positive (PPV) or negative predictive values (NPV) for in vivo protection have yet been determined.

## 3. Virus-Neutralization Assays: The Gold Standard?

In vitro virus-neutralization assays can be based on “live” virus or on pseudotyped virus-infecting permissive cells which are cultured in vitro. As indicated in Figure 1, cultured cells are infected with SARS-CoV-2 or pseudo-typed viruses expressing the S protein. Depending on the presence of neutralizing antibodies preventing infection, a reduction in plaque forming and immunostaining of SARS-CoV-2 proteins can be noticed for SARS-CoV-2, or in the case of pseudo-typed viruses, a reduction in reporter gene expression can be observed. Although virus-neutralization tests mimic infection very closely and are considered the gold standard, one has to bear in mind that the results obtained may depend on several factors, and therefore may vary from experiment to experiment, or between laboratories which perform such assays. These factors include variations of reagents for cell culture; the titer and ability of the virus stock to infect, replicate and cause cytopathic effects; the ability of the pseudovirus batch to infect and express the reporter gene in the infected cell; and the viability as well as growth of cells and the level of expression of the virus-receptor and of structures important for infection by the cultured cells, to name a few. One can expect that there will be relatively little variation in the aforementioned factors within one set of experiments performed with the same stock of virus, pseudovirus and cells, but results may vary considerably from one to another independent experiment, especially when different batches of virus, pseudovirus and cells are used. Furthermore, variations of results may be even greater when experiments are performed in different laboratories. Accordingly, it will be necessary to control these assays using defined standards for calibration and by continuous verification of the virus and pseudovirus stocks. Specific features of live virus-neutralization tests and pseudotyped virus-neutralization tests are described below and are summarized in Figure 1 and Table 1.

### 3.1. “Live” Virus-Neutralization Test (VNT)

SARS-CoV-2, like every virus, requires a permissive host cell to replicate and to release progeny for another round of infection. If antibodies present in serum can bind the viral particle by latching onto the Sprotein in such a way that they prevent attachment to the host cell receptor(s) and cell entry, they are termed neutralizing antibodies (NAbs) [53]. NAbs predominantly target four regions of the SARS-CoV-2 S protein, namely the N-terminal domain and the RBD in the S1 subunit, and the stem helix region and the fusion peptide region in the S2 subunit [53]. By back-tracking SARS-CoV-2 NAb levels in patients with breakthrough infections [54] as well as by animal studies [55,56], it was shown that the NAb titers correlated well with protection against SARS-CoV-2 infection. Therefore, NAb determination is important for the assessment of protection against infection after COVID-19 or vaccination. Several virus-neutralization tests (VNT) with authentic (live) SARS-CoV-2 have been developed to quantitate functional antibody responses that block virus infection.

#### 3.1.1. Plaque-Reduction Neutralization Test (PRNT)

The plaque reduction neutralization test (PRNT) is considered a gold standard for measuring NAbs against SARS-CoV-2 [57,58]. In PRNT (Figure 1A), a defined dose of the virus is mixed with the heat-inactivated, serially diluted test sample containing NAbs. The mixture is incubated for 1–1.5 hours (h), and afterwards added to a monolayer of a permissive cell line (usually simian Vero or Vero E6 cells) grown in duplicates or triplicates in cell culture plates. The virus-antibody mixture can be either removed after 1–1.5 h of adsorption or left on the cells throughout the assay, but in both cases the infected cell monolayer is overlaid with a semi-solid medium and incubated for several days. This allows the formation of ‘plaques’ caused by the cytopathic effect (CPE) of the virus, where a single plaque results from the infection of a single viral particle whose progeny could infect only neighboring cells due to the semi-solid overlay. The plates are fixed and stained with crystal violet solution, and plaques are usually manually counted. PRNT typically provides NAb levels as 50% neutralization titer (NT50) or 90% neutralization titer (NT90), which represents the highest dilution of a tested sample that results in a respective 50% or 90% reduction in plaques compared to a control (i.e., virus-only) sample. PRNT does not require special equipment, but is technically demanding, has low to medium throughput (as it is performed in 6-well to 24-well plates), is difficult to scale up and has a rather long turnaround time, since the virus requires 3–5 days to form visible plaques (Table 1) [57,59].

#### 3.1.2. Focus-Reduction Neutralization Test (FRNT)

The focus-reduction neutralization test (FRNT) (Figure 1A) (Table 1) is an alternative assay to PRNT. In FRNT, immunostaining against SARS-CoV-2 N or S protein, typically performed 24 h post-infection, is used to visualize ‘foci’ of infected cells, which can be counted using computer-controlled imagers, such as ELISPOT readers [57,60,61,62]. This procedure is more effective and less error-prone compared to the manual counting performed in PRNT [57]. FRNT can be performed in 24-well plates but often is scaled up to a 96-well plate format, leading to easier automation and better cost and time effectiveness. Additionally, imager software generally contains built-in curve-fitting tools, allowing for user-friendly calculation of NT50 or NT90 from focus-count data of the virus-only control and mixes of virus and serially diluted test samples [57].

#### 3.1.3. Micro-Neutralization Test (MNT)

In contrast to the two above-described methods, the micro-neutralization test (MNT) (Figure 1A) (Table 1) is always performed in 96-well plates, enabling automation and upscaling. As in previous assays, a defined dose of the virus is mixed with a serially diluted heat-inactivated test sample, and after incubation, the mixture is added to the permissive cells. After 1–1.5 h, the inoculum may be removed and replaced by fresh medium, and the plates are incubated for additional 5–7 days. Afterwards, the presence or absence of the virus-induced CPE in each well is determined. SARS-CoV-2 NAbs are quantified as a neutralization titer (NT), where the reciprocal of the serum dilution required for complete prevention of virus-induced CPE is reported [63]. Alternatively, NT50 is reported by determining sample dilution resulting in 50% virus-neutralization, often using the Spearan–Kärber or Reed–Muench formulae [59,64,65,66]. Alternatively, MNT is evaluated by in-cell ELISA, where infected cells are fixed and immune-stained for the SARS-CoV-2 N protein, and the plate is afterwards read by a standard ELISA reader [67]. For each tested well, the neutralization capacity is calculated based on the measured absorbance of the tested well and control wells. These data are used to create a four-parameter logistic regression fit of the percent neutralization and to calculate the ID50 (50% inhibitory dilution) or NT50, respectively, that in both cases stands for the reciprocal of the dilution yielding a 50% reduction in the anti-SARS-CoV-2 N protein staining [35,44,67,68]. MNT coupled to in-cell ELISA was found to be equally sensitive to PRNT in evaluating NAbs in tested sera [69], and neutralization titers determined by MNT significantly correlated with S protein-binding antibody titers determined by standard ELISA [68,70]. Since MNT uses an authentic virus that can undergo several replication cycles, the assay can be easily adapted to measure the inhibitory effect of various antiviral drugs or inhibitors of pathways that are employed by the virus for cell entry, replication or egress [67,69,71,72,73,74,75].

#### 3.1.4. Variations of above Tests

A neutralization assay with recombinant SARS-CoV-2 that is engineered to express luminescent (nanoluciferase (NanoLuc)) or fluorescent (mNeonGreen, mCherry or Venus) reporter genes is a more sophisticated variation in FRNT and MNT [74,75,76,77,78,79]. Here, an automated readout, based on the nature of the reporter gene included in the engineered SARS-CoV-2, contributes to a substantial reduction in the time needed to perform the assay. Automated fluorescent FRNT in a 96-well format with recombinant SARS-CoV-2 expressing mNeonGreen fluorescent protein from ORF7 allowed rapid evaluation of NAbs in dozens of patient sera in less than 24 h [74], while an assay with recombinant SARS-CoV-2 expressing nanoluciferase from ORF7 reduced the assay turnaround time to 5 h [75]. Importantly, results obtained by both assays correlated well with the gold standard PRNT [74,75].

#### 3.1.5. Summary of Virus-Neutralization Tests

To conclude, VNT with authentic or recombinant SARS-CoV-2 is a very effective method to determine NAbs in tested samples. The drawback, however, is that the work with “live” viruses must be performed in biosafety level-3 (BSL-3) laboratories by well-trained personnel. Whichever assay format or readout, the assay must be optimized for a particular SARS-CoV-2 isolate or variant. For example, the Omicron BA.1 variant of concern (VOC) replicates slower in Vero cells than the ancestral (D614G) SARS-CoV-2 strain, leading to the prolongation of the MNT turnaround time by 1–2 days [70,80,81].

### 3.2. Pseudotyped Virus-Neutralization Test (pVNT)

Pseudotyped viruses are viruses or viral vectors that were produced to bear foreign viral envelope proteins instead of their native ones. They provide a relatively safe model for studying virus entry because of their inability to produce infectious progeny virus. In the pseudotyped virus-neutralization test (pVNT) (Figure 1B) (Table 1), a single-cycle, replication-defective virus, generally a lentivirus [81,82,83] or a glycoprotein (G) protein-deficient vesicular stomatitis virus (VSV-ΔG) [82,84,85,86], bearing a reporter gene, is pseudotyped with the SARS-CoV-2 S protein expressed from a separate plasmid. The resulting pseudotyped virions are mixed with the test sample containing NAbs, which reduce the infection rate and thus reporter gene readout, allowing the measurement of anti-SARS-CoV-2 NAbs under BSL-2 conditions [81,82,87]. Luminescent (NanoLuc, firefly luciferase, renilla luciferase) [66,81,85,87], fluorescent (enhanced green fluorescent protein (eGFP), mNeonGreen) [88] or dual (NanoLuc-eGFP, NeonGreen-NanoLuc) [82,83,84] reporters are effectively used in both lenti- and VSV-based pseudoviruses. However, VSV pseudotypes are advantageous over their lentiviral counterparts because rapid single-round intracellular replication of the VSV genome enables robust reporter gene expression that can be detected within a few hours of infection [82].

**Table 1 ijms-24-05352-t001:** Characteristics of cell-culture-based assays to detect SARS-CoV-2 NAbs.

Assay	Performance	Assay Turnaround Time	Throughput		Biosafety Level Required
			Number of Tested Samples/Assay	VOC Testing	
PRNT	High	3–5 days	Low	Low	BSL-3
FRNT	High	1–2 days	Medium	Medium	BSL-3
MNT (CPE scoring)	Intermediate	5–7 days	Low	Low	BSL-3
MNT (In-Cell-ELISA)	High	3–4 days	Medium	Medium	BSL-3
MNT (with recombinant SARS-CoV-2)	Intermediate to high	<1 day (excluding time for virus generation)	High	Low to medium	BSL-3
pVNT (with replication-competent chimeric virus)	Intermediate to high	<1 day (excluding time for virus production)	High	Medium	BSL-2
pVNT (with replication-incompetent pseudovirus)	Intermediate	<1 day to 3 days (excluding time for pseudovirus production)	High	High	BSL-2

Assay performance is stratified based on its correlation (R^2^) with the gold standard PRNT reported in various studies [60,74,75,82]: high (r > 0.85), intermediate (r = 0.70–0.85), and low (r < 0.70). Throughput is stratified based on the capacity of the assay: (1.) to test many samples in one set (high) or not (low) and (2.) its ability to test in parallel various isolates, such as different variants of concern (VOCs).

The SARS-CoV-2 S protein used for pseudotyping is usually C-terminally truncated (lacking the last 18–21 amino acids with a putative endoplasmic reticulum retention sequence). This leads to one-log higher titers of pseudotyped virions in comparison to pseudotyping with the full-length S protein [82,89]. Since the ‘backbone’ of the pseudovirus is always the same, pseudotyping with the S protein of any SARS-CoV-2 variant including different VOCs is relatively easy, as expression plasmids bearing various S proteins can be produced by site-directed mutagenesis or obtained from commercial sources. Therefore, pVNT offers a straightforward and effective way to test the NAb functionality against various VOCs in parallel [66,83,88,90,91,92,93].

However, single-cycle, replication-defective pseudotyped viruses do not allow for any viral spread, which could impact the information obtained by the pVNT. To overcome this obstacle, replication-competent VSV/SARS-CoV-2 chimeric viruses have been generated for usage in multicycle replication-based assays [62,82,94]. There, infectious VSV was engineered to encode the SARS-CoV-2 S protein and eGFP reporter in place of the native envelope glycoprotein (G) [62,82,94]. The replication-competent VSV/SARS-CoV-2/eGFP virus used in pVNT proved more reliable in predicting the IC50 values of the human monoclonal antibodies (mAbs) against authentic SARS-CoV-2 than pseudotyped VSV-ΔG and lentiviruses, although all three surrogate viruses performed quite well in testing the neutralizing activity of both convalescent plasma and mAbs [82].

In summary, pseudoviruses and chimeric viruses containing SARS-CoV-2 S protein are relatively safe and are reliable for the detection of Sprotein-specific NAbs. However, they fall short on recapitulating the full process of SARS-CoV-2 infection, in particular regarding mechanisms that go beyond the processes for which the S protein is responsible. For example, convalescent COVID-19 patients produce antibodies against various SARS-CoV-2 structural proteins, such as surface-expressed membrane (M) protein [95,96,97], which may contribute to the neutralizing capacity of convalescent sera, as has been shown for SARS-CoV-1 M-specific antibodies [98]. Therefore, neutralization with authentic SARS-CoV-2 may be performed to verify pVNT data.

However, VNTs and pVNTs are very cumbersome for use as a screening assay for large numbers of samples, because their major disadvantages are the laborious procedures and the need for special equipment and trained personnel (Table 1).

## 4. Molecular Interaction Assays MIAs

### 4.1. Need for Molecular Interaction Assays (MIAs)

As indicated above, VNTs are considered as a kind of gold standard to determine the virus-neutralization capacity of antibodies in subjects and thus to inform us about the likelihood that they are protected against SARS-CoV-2 infection. However, VNTs and pVNTs are laborious, time-consuming, difficult to upscale for studying large numbers of samples and only few laboratories are able to run them [12]. It has been also mentioned that results obtained by VNTs, and also pVNTs, may vary considerably due to the fact the tests need to be performed with “live” virus and cultured cells.

For some viruses, it is standard practice to check for the presence of antibodies after vaccination in healthcare workers to eventually recommend re-vaccination [99,100]. SARS-CoV-2 was a newly emerging pathogen and it was therefore of paramount importance to have tools to determine the presence of neutralizing antibodies after infection and vaccination on a large scale. Since the mere measurement of SARS-CoV-2-specific antibodies cannot inform us with certainty if a subject has developed neutralizing antibodies and VNTs were available only in few laboratories, molecular interaction assays were developed in the beginning of the pandemic, allowing us to screen large numbers of serum samples for the presence of antibodies with virus-neutralizing capacity. These assays will now be introduced and discussed regarding their modes of action.

### 4.2. Principles of Molecular Interaction Assays (MIAs)

The unifying principle of molecular interaction assays mimicking the SARS-CoV-2 infection is that they measure the binding of ACE2 to RBD and RBD-containing S protein [101]. The interaction assays thus mimic the docking of RBD to ACE2, and it is possible to study antibodies and other molecules for their ability to interfere with the RBD–ACE2 interaction. As with VNTs, where the amount of virus used in the assay influences the extent of blocking, the amount of RBD influences the magnitude of blocking in the interaction assay. High amounts of infectious virus or RBD in the respective assays will adsorb high amounts of “neutralizing” antibodies. One must therefore bear in mind that the amounts of infectious virus and RBD used in the respective assays will determine the test results. This must be considered for MIAs using different amounts of RBD and serum dilutions (Table 2). It is therefore important to compare MIAs with VNTs. In fact, a very good correlation of certain MIAs and VNTs has been reported [15,102].

### 4.3. Types of MIAs

The first two published MIAs [12,13] (Table 2) were based on the principle of an inhibition ELISA (Figure 2A). In both assays, recombinant RBD was incubated for a predefined time with human serum and subsequently allowed to react with plate-bound ACE2. When no blocking antibodies are present in the serum, RBD can bind to ACE2 which can be visualized by a simple chromogenic reaction (Figure 2A) and measured by absorbance plate readers. Antibodies which bind to RBD and can block the ACE2-receptor binding lead to a reduction or abolishment of the reaction (Figure 2A).

In the assay described by Tan and colleagues, which is commercially available from Genscript as the cPass^TM^ neutralizing antibody detection kit, RBD is directly labelled with HRP (Table 2). It is of note that this assay uses a relatively small amount of RBD so that already low amounts of neutralizing antibodies will cause a strong inhibition of the RBD–ACE2 interaction. Another point to consider is that direct labelling of RBD may eventually shield epitopes responsible for ACE2- or antibody-binding. The other assay, published by Gattinger and colleagues, uses His-tagged RBD (Table 2) (Figure 2A), which is subsequently detected with a mouse monoclonal anti-His antibody followed by a secondary HRP-labelled anti-mouse IgG_1_ antibody. This assay uses larger amounts of intact RBD, which may be the reason why it allows not only the detection of blocking antibodies but also found that certain sera contain antibodies enhancing the RBD–ACE2 binding [13]. The results of the latter assay correlated well with VNTs [15,44]. An antibody-dependent enhancement of RBD binding to ACE2 was also found by other investigators later but it is not clear if the enhancement of the RBD–ACE2 interaction caused by antibodies from certain subjects has a biological effect and may eventually be responsible for the phenomenon of the antibody-dependent enhancement of disease [103,104,105,106,107,108].

The same principle of detecting the inhibition of RBD–ACE2 binding by blocking antibodies was reported by Abe et al. (Table 2) but in contrast to the assays described before, the authors immobilized RBD, pre-incubated it with serum samples and then exposed it to soluble biotin-labelled ACE (Figure 2B) [14].

A similar competitive approach was reported by Byrnes et al. (Table 2), who immobilized biotin-labelled RBD via avidin interaction to microtiter plates. This technique was chosen with the goal of maintaining the native fold of the protein. Thereafter, they added diluted serum and ACE2-Fc simultaneously to let them compete for RBD binding. Bound serum antibodies were detected via HRP-labelled anti-Fab IgG. A loss of signal was interpreted as ACE2 binding to RBD (Figure 2C) [109].

**Table 2 ijms-24-05352-t002:** Overview of first molecular interaction assays (MIAs) to detect antibodies inhibiting the ACE2–RBD interaction.

Method	Immobilized Phase (Amount)	Liquid Phase (Amount)	Preincubation Time	Serum Dilution	Reference and Time of First Publication	Reference
Inhibition ELISA	ACE2	HRP-labelled RBD (3 ng)	30 min	1:20	Tan C.W. et al. (July 2020)	[12]
Inhibition ELISA	ACE2	His-tagged RBD (100 ng)	3 h	1:2	Gattinger P. et al. (July 2020)	[13]
Inhibition ELISA	RBD (100 ng)	Biotin-labelled ACE2 (50 ng)	1 h	1:10	Abe K.T. et al. (September 2020)	[14]
Competition ELISA	RBD	ACE2-Fc (100 nM)	Simultaneous addition of serum and ACE2	1:50	Byrnes J.R. et al. (September 2020)	[109]
Flow cytometry	RBD or S	ACE2	Simultaneous addition of serum and ACE2	1:50	Gniffke E.P. (December 2020)	[110]
Luminex	RBD and S (100 pM)	ACE2 (2 µg/mL)	Simultaneous addition of serum and ACE2	1:2000	Cameron A. et al. (January 2021)	[111]

### 4.4. Further Developments of MIAs

Later in 2020 and in the beginning of 2021, other molecular assays mimicking SARS-CoV-2 infection were reported. These assays represented more elaborate methods than the previously reported ELISAs and, for example, used flow cytometry [110] or Luminex methods [111] (Table 2) (Figure 2B,C). The latter assay was designed as a three-plex assay to detect S-, RBD- and N-specific IgG antibodies simultaneously [111]. By adding soluble ACE2 in addition to the serum, blocking antibodies were identified by a reduction in the signal (Figure 2B,C).

The importance and advantages of the use of MIAs instead of mere quantitative measurement of RBD-specific antibodies have been reported previously [112]. Although convalescent patients or vaccinated subjects had comparable concentrations of RBD-specific IgG in serum samples, their capacity to inhibit RBD–ACE binding showed relevant differences [112]. Accordingly, the measurement of blocking antibodies by MIAs only, not the mere measurement of RBD-specific antibody levels, can be informative regarding the levels of protection against further infections.

### 4.5. Areas of Application for MIAs

The MIA described by Gattinger in 2020 [13] (Table 2, Figure 2A) is based on ELISA plate-immobilized ACE2 and His-tagged RBD. The assay was developed to investigate if and to what extent COVID-19 convalescent subjects mount antibody responses capable of blocking the RBD–ACE2 interaction and thus was initially thought to serve as surrogate test to estimate protection against future SARS-CoV-2 infections. In two studies, it turned out that a considerable number of convalescent subjects failed to produce RBD-specific antibodies and that their antibodies could not fully inhibit the RBD–ACE2 interaction, although RBD non-responders could mount S- and N-specific antibody responses [13,15]. Results obtained by MIA but not the levels of S- and N-specific antibodies were highly correlated with results from VNTs in the subjects analyzed. Accordingly, the results were in agreement with data obtained by others showing that the determination of S- and in particular of N-specific antibodies is useful to confirm a previous SARS-CoV-2 infection and for epidemiological surveillance studies but not for the measurement of NAbs [39,40]. Consequently, we used the MIA developed by Gattinger to investigate the development of NAbs after vaccination in comparison with VNTs and found that the MIA allows us to determine if vaccinated subjects have developed NAbs [44]. When surveying subjects who had received the first immunization with mRNA- and vector-based vaccines registered in Austria, it turned out that approximately 60–70% of the vaccinated subjects had developed antibodies which strongly inhibited the RBD–ACE2 interaction [44]. MIAs are very simple and reliable assays and we therefore consider the screening of convalescent and vaccinated subjects for the presence of NAbs one of their highly important applications (Figure 3). The use of MIAs for the large-scale screening of populations regarding their levels of NAbs is thus suggested to create a rational basis for vaccination programs.

#### 4.5.1. MIAs for Selection of Plasma, Immunoglobulin and Antibody Preparations for Treatment

One of the early therapeutic options for the treatment of COVID-19 was the use of convalescent plasma or immunoglobulin preparations [113]. It is obvious that a screening of donors for the presence and levels of NAbs or antibodies blocking the RBD–ACE2 interaction will be very helpful to improve the efficacy of passive immunization strategies for the treatment of COVID-19, because one can select donors with high levels of NAbs for manufacturing the polyclonal antibody preparations (Figure 3).

Besides plasma and immunoglobulin preparations, monoclonal NAbs also represent a possibility for the treatment and prevention of COVID-19 (Figure 3). In fact, several SARS-CoV-2 human monoclonal NAbs became available soon during the COVID-19 pandemic and were successfully tested in clinical trials [114,115]. However, it is a disadvantage of monoclonal NAbs as compared to polyclonal plasma and immunoglobulin preparations that only a few mutations in the S protein and especially within RBD can more easily affect the protective effect of a monoclonal antibody as compared to a polyclonal antibody mix in natural antibody preparations, which targets several different epitopes. A strong decrease in the efficacy of therapeutic monoclonal NAbs was especially noted for the omicron VOCs, which showed an unexpectedly high rate of mutations as compared to previous VOCs [116,117]. Clearly VNTs, pVNTs and especially MIAs represent an attractive and easy-to-handle platform for the high-throughput screening of therapeutic monoclonal antibodies intended for treatment and prevention of COVID-19, especially for immunocompromised patients and poor responders to infection and/or vaccination (Figure 3).

#### 4.5.2. MIAs for Selecting Compounds Inhibiting the RBD–ACE2 Interaction

Besides NAbs, several compounds capable of inhibiting the RBD–ACE2 interaction have also been considered for the treatment of COVID-19 [118,119]. Indeed, ACE2 was one of the first substances suggested for the treatment of COVID-19, because ACE2 in solution potently blocks the binding of RBD to cell-bound ACE2 [120,121,122]. However, the approach of systemically administering ACE2 is hampered by several problems. First of all, ACE2 has a biological function and the administration of high doses may cause unwanted side effects. Second, it may be difficult to administer doses which are high enough to compete with high viral loads. Third, the half-life of soluble ACE2 is very low, which means that it requires multiple administrations and makes treatment difficult and cumbersome. The latter two considerations may also apply for other low molecular weight compounds [123,124]. Alternatively, one may therefore consider topical administration or approaches blocking cell-bound ACE2 with compounds to prevent RBD–ACE2 interaction. Clearly MIAs will be very useful for the large-scale screening of such inhibitory compounds as a first step towards the identification of viral-entry inhibitors (Figure 3).

#### 4.5.3. MIAs for the Preclinical Evaluation of SARS-CoV-2 Vaccines

Another important area of application for VNTs, pVNTs and MIAs is the preclinical evaluation of vaccines. For example, using MIA and VNT in a preclinical study, we found that only a vaccination with a PreS-RBD fusion protein induced high NAb titers in immunized animals and overcame RBD-non-responsiveness in comparison with a purely RBD-based vaccination [44]. Likewise, we could show that vaccination with unfolded RBD did not induce NAbs as compared to immunization with folded RBD [15]. Therefore, assays measuring NAbs will be important in the preclinical evaluation of new COVID-19 vaccines but also as surrogate tests in clinical settings for measuring the efficacy of the vaccines (Figure 3).

#### 4.5.4. MIAs for the Assessment of Protective Antibody Responses in Populations

Finally, measuring NAbs has been suggested as a parameter for protection in populations, to inform us if a given population is sufficiently protected against currently circulating or emerging VOCs. Based on this population-monitoring, rational suggestions for re-vaccination or adaptions of vaccines can be made (Figure 3). Clearly, MIAs can be scaled up most easily for the measurement of large numbers of samples at reasonable costs. In this context, it should be mentioned that approximately 20% of convalescent COVID-19 patients in Austria after the first COVID-19 wave did not mount detectable levels of RBS-specific NAbs and are therefore referred to as RBD non-responders [15]. A similar percentage of RBD-non responders were found in a small group of healthy individuals after two vaccinations [125]. When determining NAbs against RBD-hu1 and RBD-omicron, it was found that patients without B-cell targeting therapy had NAbs against the wild type (hu-1), but this was significantly decreased against omicron. However, a compelling number of patients receiving B-cell targeted therapy did not mount a sufficient RBD-hu1 or omicron-specific response after the third [8,117] or fourth vaccination [117]. Regarding surveillance, we would like to emphasize that the S protein quickly acquired amino acid mutations to escape the immune response. RBD mutations started with one in the alpha variant (B.1.1.7) and peaked with 15 in the omicron variant (B.1.1.529) [125]. With the easy and fast adaption of a molecular assay regarding the VOCs, when omicron emerged it could be shown in an Austrian population that only subjects vaccinated not two but three times had the capacity to block the binding of omicron-derived RBD to ACE2 to some extent [125]. Therefore, MIAs are important tools for the surveillance of the immune response to future SARS-CoV-2 variants (Figure 3).

### 4.6. Advantages, Opportunities, Disadvantages and Challenges of MIAs

#### 4.6.1. Advantages of MIAs

In Table 3, we discuss the advantages and opportunities as well as disadvantages and challenges of MIAs in comparison with VNTs and pVNTs. One of the advantages of MIAs is that they are based on the measurement of RBD and ACE2 as protein ligands, which allows the testing of antibodies and other compounds regarding their ability to interfere with the RBD–ACE2 interaction [13]. The assay is thus very simple and the reagents are well-defined and can be produced in large quantities. By contrast, VNTs and pVNTs require infectious virus and permissive host cells, which are very difficult to “standardize”. We have mentioned above several factors, such as infectivity, virus titers and factors influencing the growth and receptor expression on the host cells, which can cause variations in VNTs and pVNTs. On the other hand, it should not be forgotten that physicochemical and structural factors can also have important effects on the interaction between RBD and ACE2. For example, it has been shown that unfolded RBD is poorly recognized by patients’ antibodies. Therefore, a reduction or lack of proper conformation of the protein will cause reduced or no binding to ACE2 [15]. Similar aspects may be valid for ACE2 and it is thus important that both proteins are expressed not only as highly pure but also functional proteins.

It is another important advantage of MIAs that they can be performed in most laboratories with simple and commonly available equipment such as ELISA readers and FACS technology. By contrast, VNTs and pVNTs require the availability of “live” virus and trained operators able to work with infectious agents in BSL2 to BSL3 laboratories, which will be available in only a few institutions in any one country. MIAs can be performed relatively quickly and are easy to calibrate with standards (e.g., defined reference antibodies, sera or ACE2) according to commonly established laboratory methods. MIAs can be benchmarked and compared with VNTs and/or pVNTs and it has been shown that results from MIAs and VNTs correlate very well [15]. It is another important advantage that one can test by MIA not only sera, body fluids and purified antibodies, but also various ligands, as long as they are kept in a buffer system that does not affect the structural integrity of the ligands. Accordingly, MIAs are especially suited for the large-scale screening of antibodies, sera and compounds for the development of prevention and treatment strategies for COVID-19. In this context, it should be mentioned that MIAs can be quickly adopted to newly emerging VOCs as soon as the amino acid sequences of the RBD of a new VOC has been determined. It takes only days to prepare synthetic genes or to mutagenize the RBD to obtain recombinant RBDs of the new VOCs ready for testing [125]. By contrast, it is very difficult, hazardous and cumbersome to propagate infectious virus derived from new VOCs and to adapt them to culture conditions for establishing a new VNT. Last but not least, it should be mentioned that MIAs have the great advantage of allowing the upscaling of test numbers for the global screening of populations regarding the presence of NAbs. Importantly, results can be obtained quickly and the cost per assay can be kept very low. Thus MIAs would be extremely suitable for the surveillance of populations who have a risk of suffering from severe COVID-19, including healthcare workers, but even for whole populations, to inform us if the subjects are protected against SARS-CoV-2 infections.

#### 4.6.2. Disadvantages of MIAs

Despite the many advantages of MIAs, there are also possible disadvantages and challenges which should be mentioned. It is clear that MIAs do not fully resemble virus infection and replication; they only simulate the binding of the virus to its receptor ACE2. Thus MIAs can only measure whether antibodies can prevent the interaction of RBD and ACE2. In this context, it must be mentioned that epitopes located outside RBD in the S protein may also play a role in virus-neutralization [45,46,47,48,49,53,126,127]. It is thus possible that certain subjects lacking antibodies blocking the RBD–ACE2 interaction may still contain NAbs. It therefore seems reasonable to retest subjects without antibodies blocking the RBD–ACE2 interaction in MIA for NAbs by VNTs or pVNTs. It must be further borne in mind that results obtained by MIAs depend on the concentrations of ligands and inhibitors. For example, if only very low concentrations of RBD are used, even sera containing low levels of NAbs will show inhibitory activity. For MIAs as well as for VNTs and pVNTs, the concentrations of RBD and virions mimicking in vivo conditions still need to be determined.

Furthermore, it has been observed that one can notice enhanced binding of RBD to ACE2 when antibodies from convalescent and vaccinated subjects are tested by MIAs. The biological relevance of the phenomenon and its potential to indicate antibody-dependent enhancement of disease is currently unknown. Obviously, persons without NAbs are at high-risk for experiencing a new SARS-CoV-2 infection. However, neither for MIAs nor for VNTs and pVNTs have thresholds and cut-offs for NAbs been established which could be used to predict with certainty if somebody is protected against a SARS-CoV-2 infection or not.

## 5. In Vivo Models

### 5.1. Overview of In Vivo Models

In this review and perspective article, we have focused on in vitro assays measuring SARS-CoV-2 neutralizing antibodies. These assays are extremely important to screen populations for the presence of NAbs, but COVID-19 is more complex and involves different parts of the immune system beyond NAbs, such as specific cellular immunity and innate immunity. Although this is not the subject of our paper, we would like also to highlight the usefulness of animal models for studying COVID-19 and host immune responses regarding parameters which cannot be assessed by the in vitro assays described by us. Regarding in vivo models, one may consider different stages of evaluating NAbs. For example, one can test if a vaccine is capable of inducing NAbs by immunizing animals with SARS-CoV-2 vaccine candidates. These in vivo models do not necessarily need to reflect disease pathology, because the endpoint is to evaluate the virus-neutralizing capacity of the induced antibodies, for example, in assays measuring SARS-CoV-2 neutralizing antibodies. In this context, one may pay attention to the fact that immunization may be carried out in inbred animals, for example, given mouse strains, or in outbred animals, such as rabbits. The advantage of immunizing outbred animals is that one can appreciate variation in the development of NAbs (e.g., responders versus non-responders) depending on genetic factors, whereas this cannot be achieved in inbred animals [44]. The ability of passive immunization with antibodies or by induction via immunization regarding protection against SARS-CoV-2 infections can be tested in several animal models which mimic symptoms and pathology of COVID-19.

### 5.2. Animal Models Mimicking Human SARS-CoV-2 Infection

Currently used animal models for SARS-CoV-2 infection are mainly hamsters, non-human primates, ferrets, cats and mice. Regarding mice, it is important to create mouse strains expressing human ACE2 to render mice susceptible for infection. The animal models show variability regarding susceptibility to different virus strains, severity of infection, symptoms and pulmonary pathology [127,128,129,130,131,132]. Currently the Syrian hamster model is one of the most frequently used animal models for SARS-CoV-2 infections [128,129].

### 5.3. The Syrian Hamster Model

In fact, the Syrian hamster has been used as a suitable animal model for a large variety of viruses, e.g., Ebola virus, West Nile virus and SARS-CoV [133]. Compared to other small animals, the immune response in Syrian hamsters seem to be physiologically similar to humans. This was of major importance in the beginning of the COVID-19 pandemic, since it was found that ACE2 expressed in Syrian hamsters can functionally interact with the SARS-CoV-2 RBD, and the animals showed high susceptibility to the virus, permitting infections with 10^3^ to 10^5^ TCID50 [118]. Importantly, Syrian hamsters show clinical symptoms 3 to 5 days after infection with SARS-CoV-2, ranging from weight loss to pathological changes in the lower respiratory tract to pneumonia, including viral shedding for up to 10 days [134,135,136]. On the other hand, comorbidities and risk factors, which were reported to be associated with severe disease outcomes in humans, cannot be mimicked in this animal model. Nevertheless, Syrian hamsters are a suitable model for pre-clinical studies in order to investigate novel forms of treatment or for in vivo evaluation of vaccine candidates for SARS-CoV-2.

Of course it must be considered that the use of animal models is connected with ethical considerations and regulatory guidelines regarding the care and use of laboratory animals. In the case of infectious diseases, especially SARS-CoV-2 and live virus-challenge experiments, animal facilities with access to BSL-3 are a prerequisite. In fact, there is a great need for such facilities and access to the Syrian hamster model is of great importance for the development of treatments, for example the novel siRNA treatment strategy for COVID-19 [137,138] and vaccines for SARS-CoV-2. The in vitro assays, like MIAs and VNTs, are important tools for the large-scale prescreening of novel therapeutics and vaccine candidates, before advancing with the development towards animal models.

## 6. Clinical Significance of Assays Measuring SARS-CoV-2 Neutralizing Antibodies

In vitro assays for measuring SARS-CoV-2 neutralizing antibodies are of high clinical relevance regarding the early accelerated screening of treatments which are centered around the induction of antibodies by vaccination, as well as of compounds which have virus-neutralizing activity. These assays are fast, relatively easy to perform and can be scaled up for the analysis of a large number of samples. If thresholds correlating with in vivo protection can established, these assays will be informative about the level of protection in populations and individuals and may pave the road to precision medicine approaches for the prevention and treatment of COVID-19 [29]. However, it is clear that these assays cannot measure the importance and contributions of cellular and innate immune responses elicited, for example, by vaccines. For example, there is evidence that it may be beneficial to target the nucleocapsid protein by vaccination [42,139,140,141]. Assays measuring SARS-CoV-neutralizing antibodies will also not inform us about innate immune defenses against SARS-CoV-2, which were found to be enhanced by genetic vaccines [142,143,144] and may be enhanced by adjuvants [145,146].

## 7. Conclusions and Suggestions

The COVID-19 pandemic seems to have stabilized currently. Although we continue to see an increase in infections, the number of COVID-19-associated deaths is relatively low in Europe. The following factors may be responsible for this. First, the majority of people have experienced a SARS-CoV-2 infection and/or have been vaccinated. Second, the currently prevailing SARS-CoV-2 strains seem to cause milder forms of disease [147] as compared to the initial variants and/or due to preexisting immunity obtained from previous infections or vaccinations.

### 7.1. Conclusions

Two things have been learned from the pandemic: neither infection nor vaccination induces sufficient amounts of NAbs, which protect completely from new infections, and the levels of NAbs decrease relatively quickly [29]. Additionally, immunocompromised patients do not fully respond to the available vaccines. Therefore, SARS-CoV-2 resembles very much the features of other respiratory viruses because infection waves can be observed every year [19]. SARS-CoV-2 is a virus which is prone to high mutation rates and it is therefore possible that new VOCs will appear which may again cause severe forms of disease. To be prepared for this, it is therefore important to continue to monitor the emergence of new VOCs and to take rapid measures, which may again include containment and the fast development of vaccines which are effective for the new VOCs. Omicron has taught us that the first generation of vaccines were much less effective against omicron [125] and accordingly there is a need for new vaccines which can be used for regular booster injections against newly emerging VOCs. There is an urgent need for the administration of booster vaccinations to be guided rationally [148]. This could be done by determination of NAb results obtained from MIAs/VNTs. We argue that a monitoring of NAbs in a population is very useful for guiding vaccination programs and especially to identify non-responders who seem to be frequent among elderly persons, ill persons and immune-compromised subjects [117]. For the latter persons, new vaccines which can overcome non-responsiveness, passive immunization and suitable medication must be quickly provided. In particular MIAs, which can be used and are affordable for large-scale screening programs, seem to be very well suited for mass screening. The broad screening for NAbs would possibly give important information to identify cut-off levels, which might be beneficial for future recommendations regarding booster vaccinations. Furthermore, the safety and efficacy of repeated booster vaccinations, with authorized vaccines as well as vaccines currently in clinical trials, have to be investigated in order to obtain high levels of NAbs, which may even allow us to come close to a condition of sterilizing immunity against SARS-CoV-2 in a population. In order to establish MIAs for mass screening, it will be necessary to define in vitro cut-off levels of NAbs, which predict in vivo protection. Additionally, negative MIA results could be re-analyzed with VNTs or pVNTs to identify NAbs not targeting the RBD. In fact, it does not seem to be too difficult to provide the scientific ground for such cut-off levels because it would seem sufficient to collect representative numbers of serum samples from persons who experienced new SARS-CoV-2 infections at the time of infection. Once a large number of such sera has been obtained together with information on whether and when they were infected, NAb levels could be determined by MIAs. Hence, it should be possible to correlate confirmed infections and MIA results to evaluate which NAb levels are too low to convey protection based on these results to define cut-off levels for protection.

### 7.2. Suggestions

We argue that research towards this direction would be very important to render MIAs useful for the prediction of protection against COVID-19 on a population basis. The monitoring of vaccine effectiveness (VE) should be a priority for epidemiological research throughout the future COVID-19 pandemic. It is well established that VE against infection declines with the emergence of new SARS-CoV-2 variants of concern (VOCs) and structured programs are therefore required [149].

COVID-19 has shown that worldwide research could advance with an unforeseen speed regarding the development of diagnostic tests and vaccines and we argue that this momentum should be kept up to achieve eventually for COVID-19 the goal of greatly reducing or even eradicating the disease, as a paradigmatic example for other respiratory virus infections [150,151] such as influenza, RSV and RV infections.

## Figures and Tables

**Figure 1 ijms-24-05352-f001:**
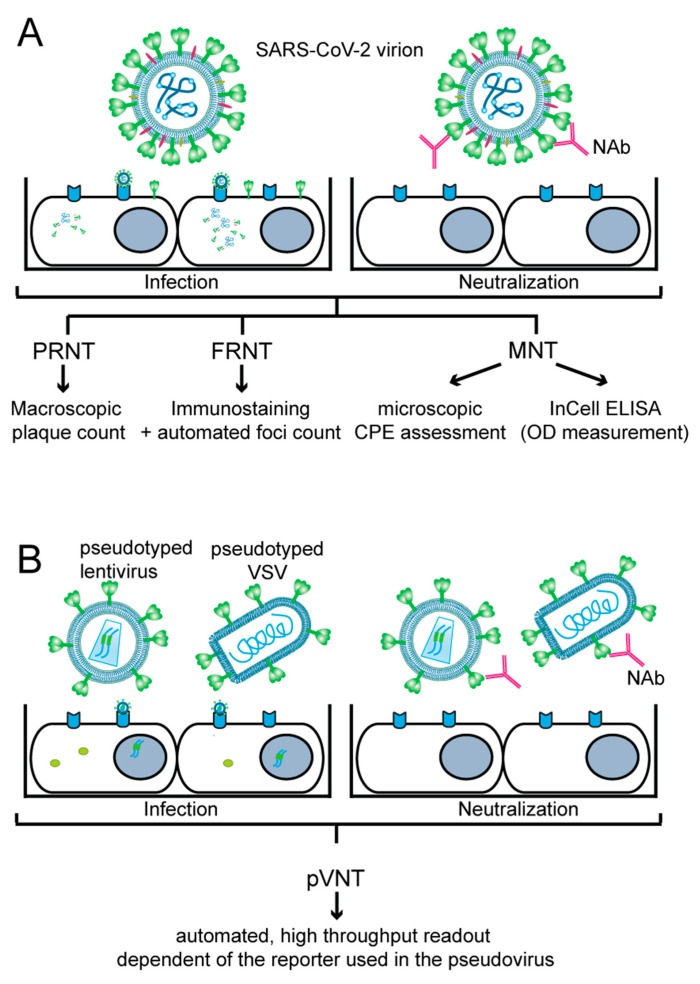
Principle of cell-culture-based neutralization assays. (**A**) Conventional virus-neutralization test (VNT) and (**B**) pseudovirus-neutralization test (pVNT) are laboratory methods primarily used for identifying whether neutralizing antibodies (NAbs) against SARS-CoV-2 are present in the patient sample. Both tests involve incubating patient serum with the virus (**A**) or pseudovirus (**B**), and then inoculating onto cell culture. If NAbs are present and the virus is neutralized, the cells will not be infected. Accordingly, there will be a reduction in plaque formation or expression of virus proteins in the plaque-reduction neutralization assay (PRNT) and focus-reduction neutralization assay (FRNT), respectively. If NAbs are not present, the virus will infect the cells. In pVNT, a recombinant or chimeric virus with SARS-CoV-2 S protein on its surface bearing a reporter gene is used for increased safety and easier readout. A reduction in the reporter gene expression reflects the presence of NAbs.

**Figure 2 ijms-24-05352-f002:**
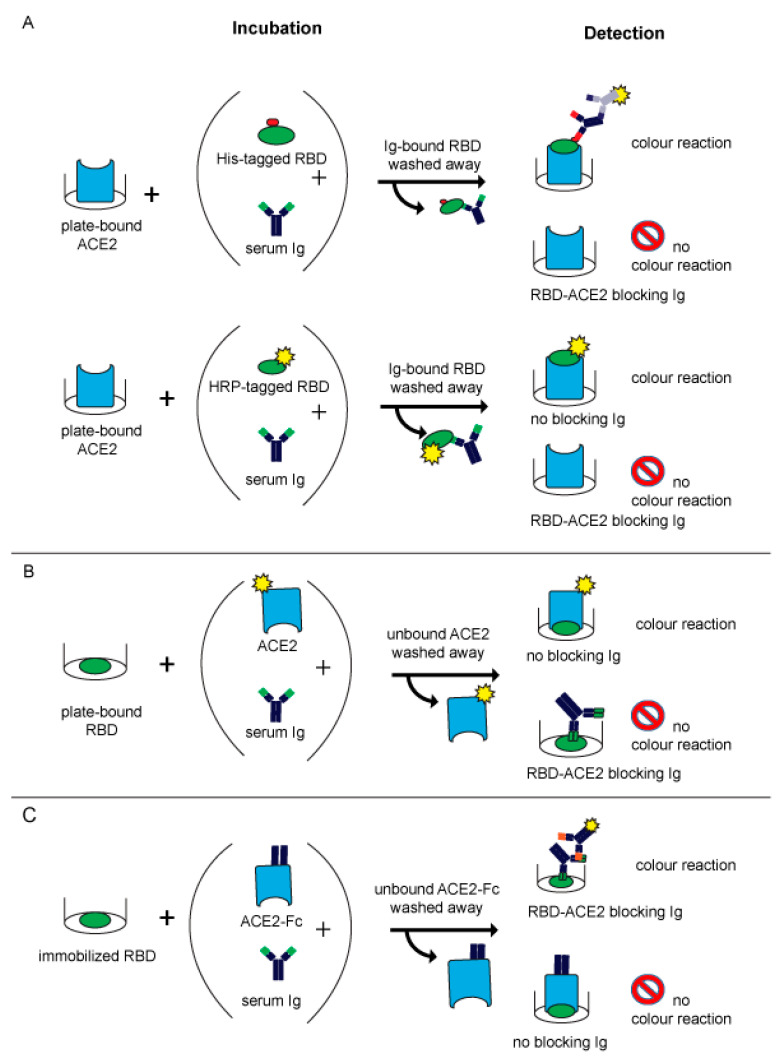
Schematic overview of test principles used in molecular assays mimicking SARS-CoV-2 infections. There are basically two types of assays measuring the interaction of RBD and ACE2 and the ability of antibodies to prevent this interaction. As shown in (**A**), one possibility is that ACE2 is immobilized and differently labelled forms of RBD are incubated with serum. The other possibility as shown in (**B**) is that immobilized RBD is incubated with a mix of soluble ACE2 and serum. Blocking antibodies are detected as a loss or reduction in signal caused by the reduced binding of labelled ACE2 to RBD. ELISA-based competition assays as shown in (**C**) use immobilized RBD. After simultaneously adding ACE2 and serum, blocking antibodies competing with ACE2 for binding to RBD are detected via colorimetric reactions.

**Figure 3 ijms-24-05352-f003:**
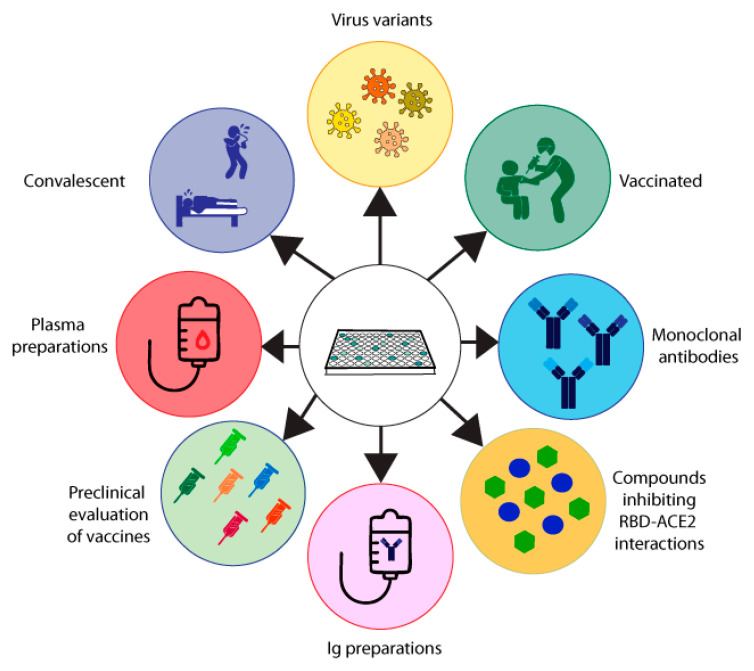
Overview of areas of application for molecular assays. Results from molecular assays mimicking SARS-CoV-2 infections can be used to determine protective antibodies of convalescent and/or vaccinated subjects or populations, as well as their blocking efficacy for emerging variants of concern. Plasma preparations and monoclonal antibodies used for preventive treatment can be tested for their neutralizing capacity.

**Table 3 ijms-24-05352-t003:** Advantages and challenges of MIAs.

Advantages and Opportunities	Disadvantages and Challenges
Simple, based on protein interaction	Do not fully resemble virus infection and replication
Reagents are well-defined	Results depend on concentration of ligands and inhibitors
Assays can be performed without the need for infectious virus	Biological relevance of enhancement is unknown
Fast and easy to standardize	Thresholds and cut-offs for in vivo protection not yes established
Correlate well with VNTs and pVNTs	
Can be used for sera, antibodies and compounds	
Easy to adapt for new VOCs	
Suitable for large-scale measurements	

## Data Availability

The data that support the findings of this study are available from the corresponding author upon reasonable request.

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
