# Peer review of "Importance, Applications and Features of Assays Measuring SARS-CoV-2 Neutralizing Antibodies"

_ijms, 2023, doi:10.3390/ijms24065352_

Round 1

Reviewer 1 Report

This is an interesting and useful review that settles a lot of arguments raised during the last two years.

I am supportive of publication after modification as indicated herebelow.

1. Please explain more thoroughly the tests. As they are now, they are over the top for the average reader.

2. Please make greater use of sub-sections and sub-sub-sections throughout the manuscript. The long passages in the manuscript are tiring.

3. Please add a new passage to underline the clinical significance of the assays.

4. Some recent significant references are missing (this can be understood, the relevant literature is advancing very fast.

After the above will have been modified, the manuscript should be re-assessed for suitability.

Author Response

This is an interesting and useful review that settles a lot of arguments raised during the last two years.

I am supportive of publication after modification as indicated herebelow.

  1. Please explain more thoroughly the tests. As they are now, they are over the top for the average reader.

Reply: We thank the reviewer for the useful comment and described the tests more thoroughly. See page 4 of the revised manuscript.

  1. Please make greater use of sub-sections and sub-sub-sections throughout the manuscript. The long passages in the manuscript are tiring.

Reply: Following the reviewers suggestion we introduced more sub-sections and sub-sub-sections. See pages 6-18 of the revised manuscript.

  1. Please add a new passage to underline the clinical significance of the assays.

Reply: Following the reviewers suggestion a new passage underlining the clinical significance of the assays was introduced. See pages 17-18 of the revised manuscript.

  1. Some recent significant references are missing (this can be understood, the relevant literature is advancing very fast.

Reply: We thank the reviewer for this comment and added additional recent and important references. Reviewer 3 has asked for expanding the section regarding animal models for SARS-CoV-2. Regarding this topic we found two very recent and comprehensive overviews regarding the animal models:

Animal Models, Zoonotic Reservoirs, and Cross-Species Transmission of Emerging Human-Infecting Coronaviruses. Kane Y, Wong G, Gao GF. Annu Rev Anim Biosci. 2023 Feb 15;11:1-31. doi: 10.1146/annurev-animal-020420-025011

Animal Models to Test SARS-CoV-2 Vaccines: Which Ones Are in Use and Future Expectations. Gimenes Lima G, Portilho AI, De Gaspari E. Pathogens. 2022 Dec 23;12(1):20. doi: 10.3390/pathogens12010020

Furthermore we identified three articles highlighting interesting aspects of non-human primate, mouse and ferret models:

Eliminating Potential Effects of Other Infections during Selection of Nonhuman Primates for COVID-19 Research. Andrade MC, Lemos BR, Silva LM, Pecotte JK. Comp Med. 2023 Jan 4. doi: 10.30802/AALAS-CM-21-000086. Online ahead of print

Generation and Characterization of a SARS-CoV-2-Susceptible Mouse Model Using Adeno-Associated Virus (AAV6.2FF)-Mediated Respiratory Delivery of the Human ACE2 Gene. Tailor N, Warner BM, Griffin BD, Tierney K, Moffat E, Frost K, Vendramelli R, Leung A, Willman M, Thomas SP, Pei Y, Booth SA, Embury-Hyatt C, Wootton SK, Kobasa D. Viruses. 2022 Dec 28;15(1):85. doi: 10.3390/v15010085

Infectious droplet exposure is an inefficient route for SARS-CoV-2 infection in the ferret model. James J, Byrne AMP, Goharriz H, Golding M, Cuesta JMA, Mollett BC, Shipley R, M McElhinney L, Fooks AR, Brookes SM. J Gen Virol. 2022 Nov;103(11). doi: 10.1099/jgv.0.001799

Reviewer 2 has asked us to mention vaccines(vaccine combinations that enhance cellular and innate immune responses. We found some interesting recent studies regarding this topic:

 SARS-CoV-2 nucleocapsid: Biological functions and implication for disease diagnosis and vaccine design. Maghsood F, Ghorbani A, Yadegari H, Golsaz-Shirazi F, Amiri MM, Shokri F. Rev Med Virol. 2023 Feb 15:e2431. doi: 10.1002/rmv.2431. Online ahead of print

An Immunological Review of SARS-CoV-2 Infection and Vaccine Serology: Innate and Adaptive Responses to mRNA, Adenovirus, Inactivated and Protein Subunit Vaccines. Al-Sheboul SA, Brown B, Shboul Y, Fricke I, Imarogbe C, Alzoubi KH. Vaccines (Basel). 2022 Dec 26;11(1):51. doi: 10.3390/vaccines11010051

DNA Oligonucleotides as Antivirals and Vaccine Constituents against SARS Coronaviruses: A Prospective Tool for Immune System Tuning. Oberemok VV, Andreeva OA, Alieva EE. Int J Mol Sci. 2023 Jan 13;24(2):1553. doi: 10.3390/ijms24021553

Innate immune mechanisms of mRNA vaccines. Verbeke R, Hogan MJ, Loré K, Pardi N. Immunity. 2022 Nov 8;55(11):1993-2005. doi: 10.1016/j.immuni.2022.10.014

Consecutive BNT162b2 mRNA vaccination induces short-term epigenetic memory in innate immune cells. Yamaguchi Y, Kato Y, Edahiro R, Søndergaard JN, Murakami T, Amiya S, Nameki S, Yoshimine Y, Morita T, Takeshima Y, Sakakibara S, Naito Y, Motooka D, Liu YC, Shirai Y, Okita Y, Fujimoto J, Hirata H, Takeda Y, Wing JB, Okuzaki D, Okada Y, Kumanogoh A. JCI Insight. 2022 Nov 22;7(22):e163347. doi: 10.1172/jci.insight.163347

Cell surface SARS-CoV-2 nucleocapsid protein modulates innate and adaptive immunity. López-Muñoz AD, Kosik I, Holly J, Yewdell JW. Sci Adv. 2022 Aug 5;8(31):eabp9770. doi: 10.1126/sciadv.abp9770. Epub 2022 Aug 3

SARS-CoV-2 host-shutoff impacts innate NK cell functions, but antibody-dependent NK activity is strongly activated through non-spike antibodies. Fielding CA, Sabberwal P, Williamson JC, Greenwood EJD, Crozier TWM, Zelek W, Seow J, Graham C, Huettner I, Edgeworth JD, Price DA, Morgan PB, Ladell K, Eberl M, Humphreys IR, Merrick B, Doores K, Wilson SJ, Lehner PJ, Wang ECY, Stanton RJ. Elife. 2022 May 19;11:e74489. doi: 10.7554/eLife.74489

Natural killer cell-mediated ADCC in SARS-CoV-2-infected individuals and vaccine recipients. Hagemann K, Riecken K, Jung JM, Hildebrandt H, Menzel S, Bunders MJ, Fehse B, Koch-Nolte F, Heinrich F, Peine S, Schulze Zur Wiesch J, Brehm TT, Addo MM, Lütgehetmann M, Altfeld M. Eur J Immunol. 2022 Aug;52(8):1297-1307. doi: 10.1002/eji.202149470. Epub 2022 Apr 22

Finally we identified new studies regarding the effects of vaccines, the interaction of RBD and ACE2 in virus variants, neutralizing epitopes outside RBD and assays analyzing neutralizing antibodies.

A Comprehensive Review of mRNA Vaccines. Gote V, Bolla PK, Kommineni N, Butreddy A, Nukala PK, Palakurthi SS, Khan W. Int J Mol Sci. 2023 Jan 31;24(3):2700. doi: 10.3390/ijms24032700

This paper provides an overview of mRNA vaccines and their future development.

COVID-19 Vaccines, Effectiveness, and Immune Responses. Abufares HI, Oyoun Alsoud L, Alqudah MAY, Shara M, Soares NC, Alzoubi KH, El-Huneidi W, Bustanji Y, Soliman SSM, Semreen MH. Int J Mol Sci. 2022 Dec 6;23(23):15415. doi: 10.3390/ijms232315415

This paper suggests that vaccination can be performed in a subject-tailored form taking into account the subjects health status. 

Revealing the Molecular Interactions between Human ACE2 and the Receptor Binding Domain of the SARS-CoV-2 Wild-Type, Alpha and Delta Variants. Hognon C, Bignon E, Monari A, Marazzi M, Garcia-Iriepa C. Int J Mol Sci. 2023 Jan 28;24(3):2517. doi: 10.3390/ijms24032517

This paper analyzes by using extended all-atom molecular dynamic simulations complemented with machine learning post-processing differences in the RBD-ACE2 interaction between different SARS-CoV-2 strains.

Cia, G.; Pucci, F.; Rooman, M. Analysis of the Neutralizing Activity of Antibodies Targeting Open or Closed SARS-CoV-2 Spike Protein Conformations. Int. J. Mol. Sci. 2022, 23, 2078

This paper identifies neutralizing antibodies which lock the pre-fusion spike protein conformation. 

Bonifacio, M.A.; Laterza, R.; Vinella, A.; Schirinzi, A.; Defilippis, M.; Di Serio, F.; Ostuni, A.; Fasanella, A.; Mariggiò, M.A. Correlation between In Vitro Neutralization Assay and Serological Tests for Protective Antibodies Detection. Int. J. Mol. Sci. 2022, 23, 9566

This paper identifies a serological test which correlates well with plaque reduction neutralization tests.

Wang, H.-I.; Chuang, Z.-S.; Kao, Y.-T.; Lin, Y.-L.; Liang, J.-J.; Liao, C.-C.; Liao, C.-L.; Lai, M.M.C.; Yu, C.-Y. Small Structural Proteins E and M Render the SARS-CoV-2 Pseudovirus More Infectious and Reveal the Phenotype of Natural Viral Variants. Int. J. Mol. Sci. 2021, 22, 908

This paper shows that a pseudovirus expressing spike, envelope and membrane proteins of SARS-CoV-2 is a useful tool to evaluate viral infectivity and testing for neutralizing activity against SARS-CoV-2 variants.

We hope that the aforementioned studies are significant and that their inclusion has helped to keep better track with the rapid emergence of literature in the SARS-CoV-2 field.

After the above will have been modified, the manuscript should be re-assessed for suitability.

We thank the reviewer and hope to have met the criteria for suitability for publication.

Reviewer 2 Report

Gattinger et al., have submitted the review entitled "Importance, applications and features of assays measuring 2 SARS-CoV-2 neutralizing antibodies."

The authors have done a wonderful job, summarizing the list of available tools to conduct neutralizing antibody assays. 

Minor suggestions

I would recommend the schematic illustrations for each assay, if possible. This would attract more readers. 

In section 5, the authors have given great thought to cellular and innate immunity. I would recommend the authors discuss a brief paragraph about the combination of vaccines that enhances cellular and innate immune response. 

Author Response

Gattinger et al., have submitted the review entitled "Importance, applications and features of assays measuring 2 SARS-CoV-2 neutralizing antibodies."

The authors have done a wonderful job, summarizing the list of available tools to conduct neutralizing antibody assays. 

Reply: We thank the reviewer for the kind comment. 

Minor suggestions

I would recommend the schematic illustrations for each assay, if possible. This would attract more readers. 

Reply: We appreciate the reviewers comment. However, Figure 1 and Figure 2 provide already schematic illustrations for each assay. In order to increase the readability of the figures we have magnified and split them. The A and B part of Figure 1 have now been put above each other and the A and B part of Figure 2 have been enlarged. 

In section 5, the authors have given great thought to cellular and innate immunity. I would recommend the authors discuss a brief paragraph about the combination of vaccines that enhances cellular and innate immune response. 

Reply: We thank the reviewer for this good comment. Indeed several approaches and considerations have been made to enhance cellular and innate immune responses by vaccines. One approach is to consider vaccines targeting proteins other than S to activate cellular immunity through antibody-mediated cellular cytotoxicity:  

SARS-CoV-2 nucleocapsid: Biological functions and implication for disease diagnosis and vaccine design. Maghsood F, Ghorbani A, Yadegari H, Golsaz-Shirazi F, Amiri MM, Shokri F. Rev Med Virol. 2023 Feb 15:e2431. doi: 10.1002/rmv.2431. Online ahead of print

Cell surface SARS-CoV-2 nucleocapsid protein modulates innate and adaptive immunity. López-Muñoz AD, Kosik I, Holly J, Yewdell JW. Sci Adv. 2022 Aug 5;8(31):eabp9770. doi: 10.1126/sciadv.abp9770. Epub 2022 Aug 3

SARS-CoV-2 host-shutoff impacts innate NK cell functions, but antibody-dependent NK activity is strongly activated through non-spike antibodies. Fielding CA, Sabberwal P, Williamson JC, Greenwood EJD, Crozier TWM, Zelek W, Seow J, Graham C, Huettner I, Edgeworth JD, Price DA, Morgan PB, Ladell K, Eberl M, Humphreys IR, Merrick B, Doores K, Wilson SJ, Lehner PJ, Wang ECY, Stanton RJ. Elife. 2022 May 19;11:e74489. doi: 10.7554/eLife.74489

Natural killer cell-mediated ADCC in SARS-CoV-2-infected individuals and vaccine recipients. Hagemann K, Riecken K, Jung JM, Hildebrandt H, Menzel S, Bunders MJ, Fehse B, Koch-Nolte F, Heinrich F, Peine S, Schulze Zur Wiesch J, Brehm TT, Addo MM, Lütgehetmann M, Altfeld M. Eur J Immunol. 2022 Aug;52(8):1297-1307. doi: 10.1002/eji.202149470. Epub 2022 Apr 22

There is also information that licensed genetic vaccines may activate the innate immune system.

An Immunological Review of SARS-CoV-2 Infection and Vaccine Serology: Innate and Adaptive Responses to mRNA, Adenovirus, Inactivated and Protein Subunit Vaccines. Al-Sheboul SA, Brown B, Shboul Y, Fricke I, Imarogbe C, Alzoubi KH. Vaccines (Basel). 2022 Dec 26;11(1):51. doi: 10.3390/vaccines11010051

Innate immune mechanisms of mRNA vaccines. Verbeke R, Hogan MJ, Loré K, Pardi N. Immunity. 2022 Nov 8;55(11):1993-2005. doi: 10.1016/j.immuni.2022.10.014

Consecutive BNT162b2 mRNA vaccination induces short-term epigenetic memory in innate immune cells. Yamaguchi Y, Kato Y, Edahiro R, Søndergaard JN, Murakami T, Amiya S, Nameki S, Yoshimine Y, Morita T, Takeshima Y, Sakakibara S, Naito Y, Motooka D, Liu YC, Shirai Y, Okita Y, Fujimoto J, Hirata H, Takeda Y, Wing JB, Okuzaki D, Okada Y, Kumanogoh A. JCI Insight. 2022 Nov 22;7(22):e163347. doi: 10.1172/jci.insight.163347

Furthermore, different adjuvant strategies are considered for activation of innate immune responses.

An adjuvant strategy enabled by modulation of the physical properties of microbial ligands expands antigen immunogenicity. Borriello F, Poli V, Shrock E, Spreafico R, Liu X, Pishesha N, Carpenet C, Chou J, Di Gioia M, McGrath ME, Dillen CA, Barrett NA, Lacanfora L, Franco ME, Marongiu L, Iwakura Y, Pucci F, Kruppa MD, Ma Z, Lowman DW, Ensley HE, Nanishi E, Saito Y, O'Meara TR, Seo HS, Dhe-Paganon S, Dowling DJ, Frieman M, Elledge SJ, Levy O, Irvine DJ, Ploegh HL, Williams DL, Zanoni I. Cell. 2022 Feb 17;185(4):614-629.e21. doi: 10.1016/j.cell.2022.01.009. Epub 2022 Feb 10

DNA Oligonucleotides as Antivirals and Vaccine Constituents against SARS Coronaviruses: A Prospective Tool for Immune System Tuning. Oberemok VV, Andreeva OA, Alieva EE. Int J Mol Sci. 2023 Jan 13;24(2):1553. doi: 10.3390/ijms24021553

We have added these aspects in the revised manuscript, see pages 17-18 of the revised version of the manuscript.

Reviewer 3 Report

The followings are some concerns and comments have been pointed out that the authors may want to consider.

1) Lines 81-82: Please include references.

2) Lines 97-104: I’d suggest the authors list the questions in order and answer them accordingly to make them clear.

3) Line 181: The format should be corrected.

4) Line 367 Figure 2: a) Please provide high-resolution images. b) It would be great if the authors could provide a relatively detailed figure legend.

5) Line 568: As a review manuscript, I don’t think the authors only listing one animal model is a good idea. There are lots of in vivo animal models. I’d suggest the author well organize this part.

6) Lines 599-647: This single paragraph is too long. Please reorganize.

Author Response

The followings are some concerns and comments have been pointed out that the authors may want to consider.

1) Lines 81-82: Please include references. 

Reply: We provided references in addition to the review by Edwards (reference 19).

2) Lines 97-104: I’d suggest the authors list the questions in order and answer them accordingly to make them clear.  

Reply: Following the reviewers suggestion we listed the questions and provided the answers which are available at the moment.

3) Line 181: The format should be corrected. 

Reply: We hope that we understood correctly and split the Figure in two parts, A above and B, below.

4) Line 367 Figure 2: a) Please provide high-resolution images. b) It would be great if the authors could provide a relatively detailed figure legend. 

Reply: We provided high resolution figures and enlarged them. Furthermore we expanded the figure legend.

5) Line 568: As a review manuscript, I don’t think the authors only listing one animal model is a good idea. There are lots of in vivo animal models. I’d suggest the author well organize this part. 

Reply: Following the reviewers suggestion we reorganized the part on animal models including also others and quoted very recent articles providing a nice overview of the currently available animal models and some recent considerations.

Animal Models, Zoonotic Reservoirs, and Cross-Species Transmission of Emerging Human-Infecting Coronaviruses. Kane Y, Wong G, Gao GF. Annu Rev Anim Biosci. 2023 Feb 15;11:1-31. doi: 10.1146/annurev-animal-020420-025011

Animal Models to Test SARS-CoV-2 Vaccines: Which Ones Are in Use and Future Expectations. Gimenes Lima G, Portilho AI, De Gaspari E. Pathogens. 2022 Dec 23;12(1):20. doi: 10.3390/pathogens12010020

Furthermore we identified two articles highlighting interesting aspects of non-human primate, mouse and ferret models:

Eliminating Potential Effects of Other Infections during Selection of Nonhuman Primates for COVID-19 Research. Andrade MC, Lemos BR, Silva LM, Pecotte JK. Comp Med. 2023 Jan 4. doi: 10.30802/AALAS-CM-21-000086. Online ahead of print

Generation and Characterization of a SARS-CoV-2-Susceptible Mouse Model Using Adeno-Associated Virus (AAV6.2FF)-Mediated Respiratory Delivery of the Human ACE2 Gene. Tailor N, Warner BM, Griffin BD, Tierney K, Moffat E, Frost K, Vendramelli R, Leung A, Willman M, Thomas SP, Pei Y, Booth SA, Embury-Hyatt C, Wootton SK, Kobasa D. Viruses. 2022 Dec 28;15(1):85. doi: 10.3390/v15010085

Infectious droplet exposure is an inefficient route for SARS-CoV-2 infection in the ferret model. James J, Byrne AMP, Goharriz H, Golding M, Cuesta JMA, Mollett BC, Shipley R, M McElhinney L, Fooks AR, Brookes SM. J Gen Virol. 2022 Nov;103(11). doi: 10.1099/jgv.0.001799

6) Lines 599-647: This single paragraph is too long. Please reorganize.

Reply: We have reorganized the discussion, conclusions and suggestions section.

Round 2

Reviewer 1 Report

The manuscript has been improved.

Two final points before acceptance.
1. A general polish of the language of the manuscript, correcting various errors or expressions difficult to understand throughout the manuscript.

2. Addition of one or two very recent relevant papers, so that the literature is fully updated.

Author Response

Reply: We thank the reviewer for his comment. We will choose the editing services provided by the MDPI publishing group. Additionally, we have added two references just published.

Didierlaurent AM, Lambert PH. Co-administration of COVID-19 and influenza vaccines [published online ahead of print, 2023 Feb 9]. Clin Microbiol Infect. 2023;S1198-743X(23)00056-3. doi:10.1016/j.cmi.2023.02.003

Morens DM, Taubenberger JK, Fauci AS. Rethinking next-generation vaccines for coronaviruses, influenzaviruses, and other respiratory viruses. Cell Host Microbe. 2023;31(1):146-157. doi:10.1016/j.chom.2022.11.016

Reviewer 3 Report

Thank you for the update. I do not have further comments on the manuscript except 1) word editing, 2) well organize manuscript format, and ensuring the figures are in high resolution for publication. Good luck.

Author Response

Reply: We thank the reviewer for his comment. 1) We will choose the editing services provided by the MDPI publishing group. 2) We have organized the format of the manuscript, by paragraphing and using underlining to highlight subsections. Also we have formatted figures and figure legends are on the same page.